# Towards Reliable Benchmarking: A Contamination Free, Controllable Evaluation Framework for Multi-step LLM Function Calling

**Seiji Maekawa    Jackson Hassell    Pouya Pezeshkpour    Tom Mitchell    Estevam Hruschka**
Megagon Labs
{seiji,jackson,pouya,tom,estevam}@megagon.ai

## Abstract

Existing benchmarks for tool-augmented language models (TaLMs) lack fine-grained control over task difficulty and remain vulnerable to data contamination. We present FuncBenchGen, a unified, contamination-free framework that evaluates TaLMs by generating synthetic multi-step tool-use tasks to stress-test TaLMs. The key idea is to cast tool use as traversal over a hidden function-dependency DAG where models must infer the correct sequence of calls to compute a target value. FuncBenchGen allows precise control over task difficulty (e.g., graph size, dependency depth, and distractor functions) while avoiding pretraining/test-time leakage. Our evaluation demonstrates reasoning-optimized models consistently outperform general-purpose models with GPT-5 significantly outperforming other available models. Performance declines sharply as dependency depth increases. Furthermore, connected distractors—irrelevant functions sharing type-compatible variables with relevant functions—prove especially difficult to handle. Also, strong models often make syntactically valid function calls but propagate incorrect or stale argument values across steps, revealing brittle state tracking by LLMs in multi-turn tool use. Motivated by this observation, we introduce a simple mitigation strategy that explicitly restates prior variable values to the agent at each step. Surprisingly, this lightweight change yields substantial gains across models. e.g., yielding an improvement in success rate from 62.5% to 81.3% for GPT-5.

  **Code**   https://github.com/megagonlabs/FuncBenchGen

## 1 Introduction

Large Language Models (LLMs) equipped with external tools ("tool use") have become central to complex real-world applications, as they enable interaction with external environments and access to up-to-date information (Luo et al., 2025). A growing body of work investigates different facets of this capability, including the ability to handle large collections of callable APIs (Kwak et al., 2025), multi-step function calling (Zhong et al., 2025; Song et al., 2025), long-horizon tool use (Kate et al., 2025), and broader evaluations of tool-use skills (Li et al., 2023; Patil et al., 2025; Sun et al., 2024).

Despite recent advances, progress in this field is impeded by two key issues. First, while prior studies have curated realistic API sets and constructed tasks to assess function-calling abilities (Li et al., 2023; Qin et al., 2024), these benchmarks often exhibit limited function-set diversity due to high curation cost, and they are vulnerable to data contamination from pretraining overlap and test-time web search (Han et al., 2025), as benchmark question–answer pairs may be publicly accessible. Second, existing benchmarks provide little fine-grained control over task difficulty, e.g., the number of required functions, the function dependency depth, and the presence of *irrelevant* functions with variables that are type-compatible with required functions. While some benchmarks (Patil et al., 2025; Qin et al., 2024; Kate et al., 2025) allow limited control over certain aspects of task complexity, such as the number of required functions, they lack comprehensive control across the function dependency depth and the presence/type-compatibility of irrelevant functions. These limitations reduce the generality of empirical findings and prevent us from understanding which factors most

Table 1: Comparison of previously proposed function-calling testbeds with FuncBenchGen. Func. indicates function.

| | Contamination-Free | Task Complexity (Controllability) | | |
| --- | --- | --- | --- | --- |
| | | Required Func. Size | Dependency Depth | Irrelevant Func. Type |
| API-Bank (Li et al., 2023) | ✗ | ✗ | ✗ | ✗ |
| BFCLv4 (Patil et al., 2025) | ✗ | ✓ | ✗ | ✗ |
| ToolBench (Qin et al., 2024) | ✗ | ✓ | ✗ | ✗ |
| ComplexFuncBench (Zhong et al., 2025) | ✗ | ✗ | ✗ | ✗ |
| LongFuncEval (Kate et al., 2025) | ✗ | ✓ | ✗ | ✗ |
| FuncBenchGen (ours) | ✓ | ✓ | ✓ | ✓ |

significantly impact model performance. We summarize the existing benchmarks and compare them with our proposed framework in Table 1.

In this paper, we aim to isolate and analyze the core capabilities and failure types of tool-augmented LLMs (TaLMs) in multi-step function calling scenarios. To this end, we propose FuncBenchGen, a framework for automated generation of contamination-free function calling tasks with controllable difficulty to enable systematic evaluation and analysis of TaLMs, as illustrated in Figure 1. The key idea is to represent function dependencies as a directed acyclic graph (DAG) and frame multi-step function calling as a graph traversal problem. Given a set of input variables with known values, a target variable, and a set of external function schemas, the task requires an agent to determine the value of the target variable by executing an appropriate sequence of external function calls.

Our key contributions are as follows:

- We introduce **FuncBenchGen**, a novel evaluation framework for tool-augmented language models (TaLMs). The framework automatically generates contamination-free function-calling tasks with controllable difficulty, specified by parameters such as the number of required function calls, the number and types of input/output variables, and the number and connectivity of irrelevant functions.

- Using FuncBenchGen, we conduct extensive experiments with seven state-of-the-art open and closed LLMs. Our analysis yields several important findings: reasoning-optimized models consistently outperform general-purpose ones, yet even GPT-5 struggles with longer function call sequences. For example, it achieves only a $15\%$ success rate when 20 function calls are required. We also observed that external functions that are *irrelevant* to solving the problem, but that are "connected" to the solution DAG (i.e., that share type-compatible variables with functions involved in the solution) severely degrade performance for all LLM models. Moreover, the task success is strongly influenced by graph structure, with shallower and more sequential dependency chains being easier to solve; and although most models invoke functions with correct syntax, they frequently fail to propagate argument values across calls.

- Finally, identifying key failure types in models function calling capability, we propose a simple augmentation mitigation strategy that restates variable values from prior calls, which significantly improves success rates across models—for example, yielding an improvement in success rate from 62.5% to 81.3% for GPT-5.

## 2 RELATED WORK

**Multi-step Reasoning in LLMs**   Recent LLMs demonstrate impressive multi-step reasoning capabilities across various domains, including mathematical problem solving (Davoodi et al., 2025), multi-document reasoning (Maekawa et al., 2025), and commonsense reasoning (Yu et al., 2025). Other work has focused on the rigor and logical validity of this reasoning. For example, FOLIO (Han et al., 2024) and Multi-LogiEval (Patel et al., 2024) evaluate complex multi-step logical reasoning using first-order logic. Despite these advances, the robustness of their multi-step reasoning capabilities remains unclear when models are required to interact with external tools.

**Code Generation and Reasoning**   The ability to use tools is fundamentally linked to a model's ability to reason about code execution. This foundational capability has been evaluated by recent

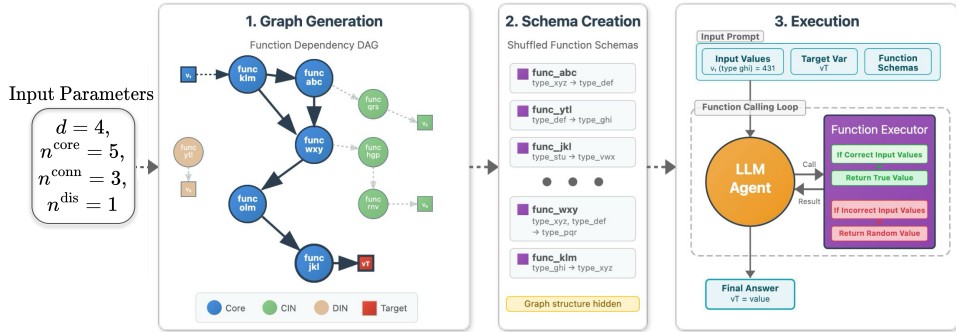

Figure 1: Overview of FuncBenchGen. $d, n^{\text{core}}, n^{\text{conn}}$, and $n^{\text{dis}}$ indicate the dependency depth and the numbers of core nodes, connected irrelevant nodes (CINs), and disconnected irrelevant nodes (DINs), respectively.

benchmarks (Gu et al., 2024; Chen et al., 2025). While these benchmarks focus on reasoning about code, our work evaluates the applied task of using code as black-box external tools.

**Function Calling in LLMs** Models such as ToolLLM (Qin et al., 2024), Gorilla (Patil et al., 2024), and ToolACE (Liu et al., 2025a) highlight the importance of equipping LLMs with access to vast API collections to tackle real-world use cases. Tool-Planner (Liu et al., 2025b), ToolDial (Shim et al., 2025), HammerBench (Wang et al., 2025) further extend these capabilities by incorporating planning and multi-turn dialogue for more sophisticated tool-use by using LLMs' multi-step reasoning capabilities. Meanwhile, LongFuncEval (Kate et al., 2025) and ComplexFuncBench (Zhong et al., 2025) focus on evaluating LLMs' ability to handle long-context and multi-step function calling scenarios. Despite these advances, the evaluation datasets used often lack generality and controllability, limiting their utility for systematic analysis.

**Contamination and Robustness in Evaluation** Robustness and contamination issues in benchmark construction pose significant challenges. Datasets such as (Mirzadeh et al., 2025; Shojaee et al., 2025) have addressed the limitations of LLM reasoning evaluation due to dataset contamination or task leakage, e.g., when the LLM training set includes a specific tool or task included in the test set. Han et al. (2025) have shown data contamination can happen during web search tool-use. White et al. (2025) have introduced LiveBench, a contamination-limited benchmark that is refreshed regularly and spans multiple domains such as math, reasoning, and instruction following tasks. Though these studies emphasize the importance of creating bias-free and contamination-free tasks, they do not specifically address the challenges in multi-step function calling scenarios. In contrast, our work seeks to bridge this gap by offering a principled, automated framework for function set and task generation, supporting controlled stress-testing and robust analysis of TaLM capabilities.

## 3 THE EVALUATION FRAMEWORK FOR FUNCTION CALLING IN LLMS

We introduce FuncBenchGen, a benchmark generation framework designed to evaluate the multi-step function calling capabilities of LLMs. FuncBenchGen automatically creates synthetic, contamination-free function sets and tasks with controllable difficulty, enabling systematic analysis of the factors that affect model performance without the confounding influence of data bias or leakage. We formalize multi-step function calling as a graph traversal problem defined over a directed acyclic graph (DAG) of function dependencies. Importantly, LLMs are given only the list of generated functions and input variables, rather than the dependency graph itself, requiring them to infer the correct call sequence. The framework overview is illustrated in Figure 1.

### 3.1 TASK DEFINITION: MULTI-STEP FUNCTION CALLING

Given a set of functions $\mathcal{F} = \{f_1, f_2, \ldots, f_n\}$, where each function $f_i$ takes a set of input variables $\mathcal{V}_i^{in}$ and produces a single output variable $v^{\text{out}}$, along with a set of input variables $\mathcal{V}_{input} = \{v_1, v_2, \ldots, v_k\}$ with known values and a target variable $v_T$, the LLM agent is tasked

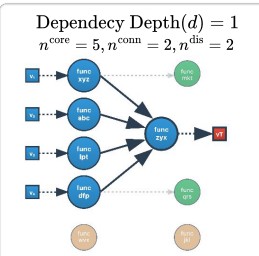 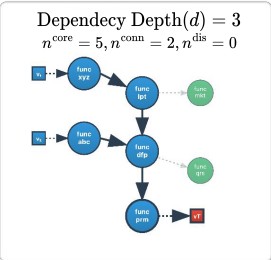 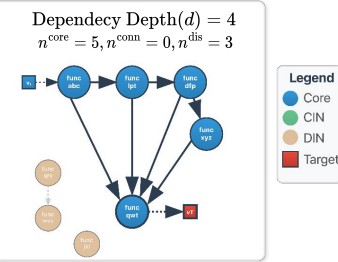

Figure 2: Examples of the different kinds of graphs that can be generated with 5 core nodes. CIN indicates connected irrelevant nodes and DIN indicates disconnected irrelevant nodes. $n^{\text{core}}$ is the number of core nodes, $n^{\text{conn}}$ is the number of CINs, and $n^{\text{dis}}$ is the number of DINs.

to determine the value of $v_T$ by iteratively executing a sequence of function calls. For each execution step, the LLM agent can call any number of functions from $\mathcal{F}$ and then the system returns function outputs based on the input values specified by the LLM agent. The process ends when no further calls are made, at which point the agent's final output is parsed to obtain the answer.

## 3.2 BENCHMARK GENERATION PROCESS

FuncBenchGen is designed based on two core principles to address limitations of existing benchmarks, as discussed in Section 1: 1) **Contamination-Free**: The framework generates synthetic function sets and tasks at evaluation time, ensuring that no pretraining or test-time leakage can occur. 2) **Controllable Task Complexity**: The framework allows us to precisely control multiple dimensions of task complexity by taking as inputs the number of functions, the function dependency depth, and the amounts and type of irrelevant functions. This enables systematic analysis of how different complexity factors affect model performance by isolating each factor.

To ensure contamination-free evaluation, FuncBenchGen generates synthetic function sets and tasks at evaluation time. Also, to control task complexity, we represent function dependencies as a directed acyclic graph (DAG), where nodes represent functions and edges represent the dependencies between functions, and formulate multi-step function calling as a graph traversal problem. This idea allows us to precisely control multiple dimensions of task complexity by manipulating the underlying graph structure. Formally, a function dependency graph $G = (\mathcal{F}, \mathcal{E})$ is a DAG where each node $f_i \in \mathcal{F}$ represents a function with input variables $\mathcal{V}_i^{in}$ and output variable $v_i^{\text{out}}$ and each directed edge $(f_i, f_j) \in \mathcal{E}$ indicates that function $f_j$ can consume the output from function $f_i$. To generate the function dependency graph $G$, the framework takes as inputs the number $n^{\text{core}}$ of functions that will be required to solve the task, the dependency depth $d$, and the number of irrelevant distractor functions that are connected to the solution DAG ($n^{\text{conn}}$) and the number of irrelevant distractor functions that are disconnected from the DAG ($n^{\text{dis}}$). Recall that we define function $f$ to be connected to the solution DAG if and only if there exists a dependency edge between $f$ and at least one function in the solution DAG. We call the connected irrelevant nodes/functions *CIN* nodes and the disconnected irrelevant nodes/functions *DIN* nodes for brevity.

The benchmark generation process consists of two main steps: 1) **Graph Structure Generation**: Given parameters specifying the desired graph characteristics that meet the target task difficulty, the framework constructs a DAG that satisfies these constraints including the number of required functions, the dependency depth, and the presence and type of irrelevant external functions. 2) **Function Schema Creation**: Each node in the dependency graph is converted into a function schema with a randomly generated name. This on-the-fly generation ensures contamination-free evaluation.

**Graph Structure Generation**   The framework takes a two-stage approach: 1) core node creation and 2) irrelevant node addition. First, we create a node sequence of length $d$ to ensure that the generated graph contains a valid path from input variables to the target variable with the required number of function calls. Then, we iteratively add the remaining $n^{\text{core}} - (d + 1)$ core nodes by randomly adding new parent nodes to the existing core nodes, while ensuring acyclicity and dependency depth. Second, we add irrelevant nodes according to the specified connection type. For connected irrelevant nodes (CINs), we randomly select nodes from the existing core nodes and add the irrelevant nodes

as their children. For disconnected irrelevant nodes (DINs), we create isolated nodes and then add at most $\lfloor n^{\text{dis}}/2 \rfloor$ edges between them while ensuring acyclicity, to simulate the possible function dependency between irrelevant nodes. We provide illustrative examples of the generated graphs and more graph generation details in Appendix A.1.

**Function Schema Creation**  For the model input, each node in the dependency graph is converted into a function schema with: (1) Function name: A randomly generated identifier (e.g., func_yep). (2) Input parameters: Variables with randomly assigned type and subtype annotations. (3) Output variable: A single variable that can serve as input to dependent functions. And, (4) Description: Natural language explanation of the function's purpose.

Functions are linked through semantic type and sub-type matching rather than variable names, serving as a lightweight proxy for the semantic reasoning that connects functions in real-world scenarios. In practice, if a function's input type and sub-type matches another's output type and sub-type, the two are connected. See B.2 for details.

## 3.3 REAL-WORLD APPLICABILITY

Many real-world function calling scenarios can be mapped directly to DAG structures. For example, a user may ask an LLM agent to find a sightseeing tour near their hotel tomorrow and reserve a taxi for them an hour after the tour ends (example from ComplexFuncBench (Zhong et al., 2025)). Here, an agent must: 1) find the hotel's location, 2) use the location to find nearby tours, 3) filter nearby tours by date and availability, and 4) reserve a taxi at the correct time and place. Figure 3 illustrates the function dependency graph for this example. This workflow naturally forms a dependency graph where each function depends on specific outputs from previous calls—tours cannot be searched for without the hotel's location, a tour cannot be selected without the tour search results, etc. Our framework abstracts this complexity into a controllable evaluation setting while preserving the essential challenge: models must infer these dependencies from function signatures alone, just as they would when interacting with unfamiliar APIs in deployment.

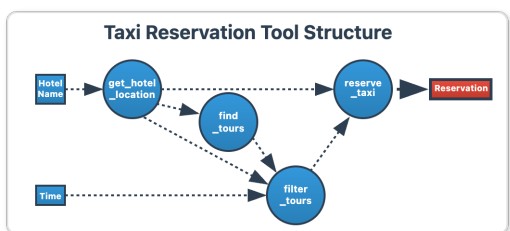

Figure 3: Graphical representation of a real tool-calling example sequence. Many tool-calling applications can be represented as DAGs in this way.

## 3.4 IMPLEMENTATION DETAILS

Each variable in the system is assigned a three digit random integer value. Functions implement deterministic logic: they return the correct output value only when provided with the exact expected input values, which the agent must discover by calling the parent functions of that node. They return random incorrect outputs when given invalid inputs, simulating realistic API behavior where invalid parameters lead to silent failures or unhelpful responses. For the sake of simplicity, we set the number of output variables for each function to one.

We implement an interaction protocol between the LLM agent and the function execution environment. At each step, the model can make multiple function calls, then receive their outputs, and continue until it makes no further calls. An example of the model input is shown in Appendix B.3.

## 4 EXPERIMENTS

We apply our FuncBenchGen framework to conduct extensive experiments to explore the following research questions: **(RQ1)** How do LLMs perform in function calling tasks as the size of the core function set varies? **(RQ2)** How do irrelevant functions affect model performance? **(RQ3)** How does the function dependency depth impact performance? **(RQ4)** How do larger function sets and thinking budgets affect performance of the best performing models? **(RQ5)** What are common failure types in function calling tasks and how can they be mitigated?

Table 2: Success rates and average number of function calls (**ACs**) made by models. Results are aggregated across all test configurations: number of irrelevant functions $\{0, 5, 10, 20\}$, irrelevant node connectivity type {Connected, Disconnected, Half-and-Half}, and graph dependency depth $\{1, ...n^{\text{core}} - 1\}$, with 5 random trials per configuration. ACs (Succ.) and (Fail.) denote the ACs for successful and failed trials, respectively. "–" indicates that there are no successful trials.

| # core nodes | 5 | | | 10 | | | 20 | | |
|---|---|---|---|---|---|---|---|---|---|
| | Success Rate | ACs (Succ.) | ACs (Fail.) | Success Rate | ACs (Succ.) | ACs (Fail.) | Success Rate | ACs (Succ.) | ACs (Fail.) |
| GPT-5 | 72.5% | 5.4 | 7.6 | 38.2% | 11.5 | 13.0 | 15.0% | 22.0 | 23.7 |
| GPT-5-mini | 16.0% | 5.0 | 7.9 | 7.6% | 10.0 | 14.2 | 4.2% | 20.0 | 23.4 |
| Gemini-2.5-Pro | 46.5% | 5.5 | 6.3 | 14.4% | 10.6 | 13.2 | 6.0% | 22.6 | 24.4 |
| Gemini-2.5-Flash | 31.5% | 5.1 | 1.1 | 13.8% | 10.3 | 1.3 | 7.2% | 20.0 | 1.3 |
| Qwen3 | 11.0% | 5.4 | 6.1 | 8.2% | 11.2 | 11.1 | 3.8% | 24.1 | 19.3 |
| GPT-4.1 | 12.0% | 5.2 | 4.1 | 2.2% | 10.3 | 7.1 | 0.2% | 21.0 | 12.0 |
| GPT-4.1-mini | 11.0% | 5.2 | 5.0 | 0.0% | – | 9.0 | 0.0% | – | 18.9 |

## 4.1 SETUP

To investigate our research questions, we test LLMs under various controlled conditions varying graph size, number of irrelevant functions, and the depth of required function call sequences. For graph size, we vary the number of core nodes in $\{5, 10, 20\}$. We set the number of added irrelevant nodes to $\{0, 10, 20, 40\}$. For each irrelevant node setting, we test three different connection types: 1) **Connected**: all irrelevant nodes are connected to the core nodes in the dependency DAG, 2) **Disconnected**: all irrelevant nodes are disconnected from the core nodes, and 3) **Half & half**: half of the irrelevant nodes are connected to the core nodes and the other half are disconnected. We also vary the dependency depth between 1 and $n^{\text{core}} - 1$ for graphs with 5 and 10 core nodes. For graphs with 20 core nodes or more, we set the dependency depth for every $10\%$ increment starting from 1, i.e., $1, 3, \ldots, 17, 19$ for 20 core nodes. To obtain reliable results, we generate 5 different graphs for each setting of core nodes, irrelevant nodes, and dependency depth and report the average results.

**Models** We evaluate various LLMs with reasoning-optimized and general-purpose capabilities. Reasoning models include both closed and open models: GPT-5, GPT-5-mini, Gemini-2.5-Pro, Gemini-2.5-Flash, Qwen3-235B22A (Qwen3 for short). General models include GPT-4.1 and GPT-4.1-mini. The model details are summarized in Appendix B.1. We set the thinking budgets of all the reasoning models to their default, e.g., medium for GPT-5, and set the temperature and top-p parameters to 0.0 and 1.0, respectively, for all our experiments.[1]

**Evaluation Metrics** We evaluate the models based on both success rate and efficiency in function calling. A function call sequence is considered correct if it produces the expected output for the task. To measure efficiency, we record the number of function calls generated by each model. Given budget constraints, we cap the maximum number of calls at twice the minimum required number. A more detailed analysis of function-calling sequences is provided in Section 4.5.

## 4.2 MAIN RESULTS (RQ1)

Table 2 shows the overall results of all models across different numbers of core nodes. The results are averaged over no extra and $10, 20, 40$ irrelevant nodes (functions) of three connectivity types with five generated graphs for each setting.

**GPT-5 outperforms all other models by a significant margin, but it reaches only 15% on extended function-call sequences, while general-purpose models fail to reliably solve even simpler tasks.** Reasoning-optimized models, e.g., GPT-5 and Gemini-2.5-Pro, consistently surpass general-purpose models (GPT-4.1, GPT-4.1-mini) at every core size. With 5 core nodes, GPT-5 attains a $72.5\%$ success rate versus $12.0\%$ for GPT-4.1. However, success rates fall sharply as sequence length grows: GPT-5 drops from $72.5\%$ (5 core nodes) to $15.0\%$ (20 core nodes), and Gemini-2.5-Pro from $46.5\%$ to $6.0\%$. These trends suggest limited effective planning depth and only modest self-correction via reflection as tasks require longer function-call sequences.

---

[1]Since GPT-5 and GPT-5-mini do not support the temperature parameter, we use their default settings.

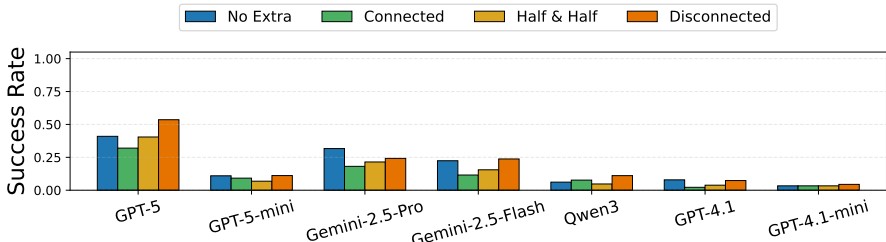

Figure 4: Success rate under different irrelevant node connection types. The results are averaged over $\{5, 10, 20\}$ numbers of core nodes and $\{0, 10, 20, 40\}$ irrelevant nodes. Connected, Half & Half, and Disconnected indicate that all, half, and none of the irrelevant nodes are connected to the core nodes, respectively.

**GPT-5 invokes 10% more calls than necessary even when successful.** While GPT-5 demonstrates strong performance, it still exhibits inefficiencies in function calling. For example, with 10 and 20 core nodes (i.e., required function calls), GPT-5 averages 11.5 and 22.0 function calls respectively for successful cases. This indicates that even the best-performing models struggle to optimize their function call sequences, often making unnecessary calls.

**The trends of average calls on failed cases depend on the model.** We observe distinct patterns in the average number of function calls (ACs) for failed cases across different models. For example, GPT-5, GPT-5-mini, Gemini-2.5-Pro tend to make more function calls in failed cases than successful ones, indicating that they struggle to find the correct path once they deviate. In contrast, Gemini-2.5-Flash invokes significantly fewer function calls in failed cases compared to successful ones, suggesting that it gave up when it could not find the correct path.

### 4.3 Effect of Irrelevant Functions (RQ2)

**Connected irrelevant nodes (CINs) severely degrade performance for all models.** Figure 4 shows that CINs have the most negative impact on model performance compared to other types across most tested LLMs. We speculate that this is because CINs share variables with core nodes, making it challenging for models to distinguish relevant function paths from irrelevant ones. The similar trend of the performance drop is observed in other reasoning tasks (Shi et al., 2023) when distracting information is present.

Interestingly, the effect of disconnected irrelevant nodes (DINs) varies across models. For instance, GPT-5 achieves better performance in the "Disconnected" setting than in the "No Extra" setting, suggesting that the existence of DINs may prompt GPT-5 to carefully consider its function calls, potentially leading to better performance in the presence of irrelevant functions. In contrast, Gemini-2.5-Pro performs worse in the "Disconnected" setting than the "No Extra" setting, suggesting that it may struggle to ignore irrelevant functions even when they are disconnected from the core nodes.

We also observe that "Half & Half" settings often yield intermediate performance between the "No Extra" and "Connected" settings, indicating that the presence of some CINs is sufficient to significantly confuse the models.

### 4.4 Effect of Function Dependency Depth (RQ3)

We break down the results by the dependency depth in Figure 5. We also show the results with $95\%$ confidence intervals and more numbers of core nodes in Appendix C.

**Lower dependency depth is more manageable.** Lower dependency depth leads to higher success rates across all models. For instance, with 10 core nodes, GPT-

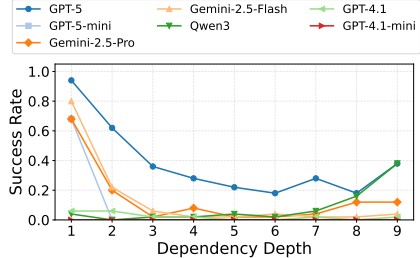

Figure 5: Success rates by the dependency depth. The number of core nodes is set to 10. The results are averaged over all irrelevant node settings.

5 achieves near a $90\%$ success rate when the dependency depth is set to 1, where a graph has a star structure (see Figure 2), but this rate drops sharply to less than $30\%$ when the dependency depth increases, i.e., between 4 and 8. This pattern is consistent across other reasoning-optimized models like Gemini-2.5-Pro and GPT-5-mini, suggesting higher dependency depth poses significant challenges for LLMs in function calling tasks.

**Fewer branches in the function call sequence are less harmful.** We observe that GPT-5, Gemini-2.5-Pro, and Qwen3 show a slight performance improvement when the dependency depth is set to 8 and 9, where a graph has a path structure, compared to dependency depth between 5 and 7. This suggests that having fewer branches in the function call sequence may be less harmful, as it reduces the complexity of decision-making at each step. However, this trend is not observed in smaller models like GPT-5-mini and Gemini-2.5-Flash, indicating that this benefit may be more pronounced in larger, more capable models.

### 4.5 DISCUSSION (RQ4)

To better understand the behaviors of the best performing model, we conduct experiments using GPT-5 with larger function sets and different thinking budgets.

**Larger sets of required nodes are generally more challenging even for the disconnected setting.** Figure 6 shows results for each number of core nodes, where results are averaged across the numbers of irrelevant nodes.

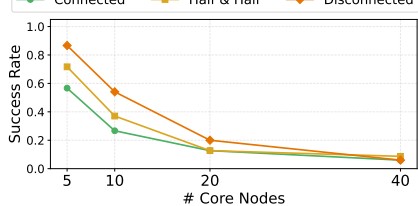

Surprisingly, even in the disconnected setting where irrelevant functions are not type-compatible with core functions, GPT-5's performance degrades significantly (lower than $10\%$ for 40 core nodes) as the number of core nodes increases. This poor performance of GPT-5 indicates that even the best models are not ready to be used over a large function sets which is happening with existing MCP servers (Anthropic, 2025).

Figure 6: Success rate of GPT-5 with various numbers of core nodes. The results are averaged across $\{0, 10, 20, 40\}$ irrelevant nodes.

**Sufficient thinking budget is necessary in complex function calling tasks.** To investigate in detail how the thinking budget affects the performance of GPT-5, we compare the medium thinking budget,[2] i.e., our default setting in other experiments, with the minimal thinking budget in Figure 7. We observe a significant drop from the medium thinking budget setting, indicating that the minimal budget severely limits the model's ability to reason through multi-step function calling tasks. Only for the no extra irrelevant nodes setting, GPT-5 with the minimal thinking budget obtains higher than $50\%$ success rate. In other settings, the success rates are less than $20\%$. This result highlights the importance of providing sufficient reasoning capacity for LLMs to effectively navigate complex function calling scenarios.

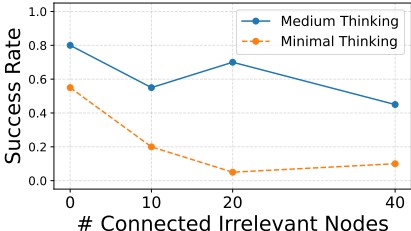

Figure 7: Thinking budget comparison for GPT-5 with 5 core nodes, averaged over all irrelevant node settings.

### 4.6 FAILURE ANALYSIS AND MITIGATION STRATEGY (RQ5)

In analyzing model failures in function calling, we use complete execution traces from our in-house executor, which logs all calls, arguments, returns, and state updates, enabling deterministic, annotation-free attribution. The executor enforces four sequential, machine-checked predicates—name resolution, schema conformance, dataflow availability, and value consistency—designed to be minimal and collectively exhaustive. We categorize the failure cases into four types: 1) **Function Not Found**: The model attempts to call a function that does not exist in

---

[2]While GPT-5 has four options for its thinking budgets, high, medium (default), low, and minimal, we fucus on the medium and minimal thinking budgets in this paper due to the budget constraint.

Table 3: Failure types. The results are averaged $\{5, 10, 20\}$ core nodes and all irrelevant node settings, with 5 trials each.

| Failure type | GPT-5 | GPT-5-mini | Gemini-2.5-Pro | Gemini-2.5-Flash | Qwen3 | GPT-4.1 | GPT-4.1-mini |
|---|---|---|---|---|---|---|---|
| Function Not Found | 0.0% | 0.0% | 2.4% | 0.0% | 0.0% | 0.0% | 0.0% |
| Wrong Number of Inputs | 0.0% | 0.0% | 0.2% | 0.9% | 0.1% | 0.0% | 0.0% |
| Value Not Yet Known | 79.6% | 66.8% | 69.1% | 81.3% | 74.0% | 73.2% | 66.8% |
| Incorrect Value | 20.4% | 33.2% | 28.3% | 17.9% | 25.8% | 26.8% | 33.2% |
| Total errors | 6,054 | 13,180 | 5,756 | 235 | 9,472 | 3,597 | 8,560 |

the provided function list. 2) **Wrong Number of Inputs**: The model provides more or fewer input arguments than the function schema allows. 3) **Value Not Yet Known**: The model tries to use a variable whose value has not been established through prior function calls or initial inputs. 4) **Incorrect Value**: The model uses a variable with an incorrect value, which has been established through prior function calls or initial inputs. These failure types cover all systematically detectable errors, as they correspond to the hierarchical prerequisites required for any valid function execution: 1) function existence errors, 2) argument mismatch errors, 3) incorrect call sequence errors, and 4) state-tracking errors. These failure types can be considered in sequence (1–4) using if-branches, and any error arising from a function call will fall into one of these categories.

**Models attempt to use unknown variable values most frequently.** Table 3 summarizes the failure types of all models, where the results are aggregated over function calls made by each model. We observe that the most common failure mode across all models is attempting to use variable values that are not yet known, accounting for over 66% of failures in every model. This indicates that models often struggle to accurately track which variables have been established through function calls. While recent models have shown strong value retrieval capabilities (Hsieh et al., 2024) from given long documents, they still face challenges in maintaining context over multiple steps in function calling scenarios.

Interestingly, most models rarely attempt to call non-existent functions or provide too many inputs, suggesting that they generally understand the function schemas provided.

### 4.6.1 MITIGATION STRATEGY

To address the most common failure types of using unknown variable values and incorrect values, we propose a simple mitigation strategy. Instead of each function just returning the value of its output variable, it also returns the list of all known variable values as well.

This provides no extra information to the model - it simply restates the values of all the variables the model has already discovered, including variables with wrong values that have been discovered through incorrect function calls. This provides the model with explicit context about which variables are available for use in function

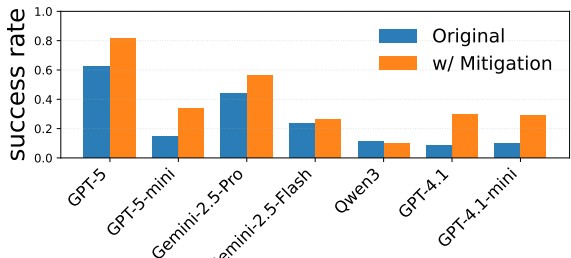

Figure 8: Comparison between the baseline and the proposed mitigation strategy. The number of core nodes is set to 5. The results are averaged across $\{0, 10, 20, 40\}$ CINs, with 5 trials each.

calls. This lightweight approach does not rely on the FuncBenchGen framework, and can easily be implemented in real-world scenarios with a simple wrapper around the provided functions.

**Simple variable reminders dramatically improve performance.** Figure 8 compares the success rates of all models with and without the above mitigation strategy. We observe that the mitigation strategy improves performance over both reasoning and general models. Gemini-2.5-Flash and Qwen3 have a small improvement or a slight performance drop, suggesting that these models may struggle to effectively utilize the additional context provided. Overall, this result indicates that even a simple reminder of known variable values can significantly enhance LLMs' function calling capabilities by reducing errors related to variable usage.

$d = 5, n^{\text{core}} = 10, n^{\text{conn}} = 10, n^{\text{dis}} = 0, seed = 0$ $\qquad$ $d = 5, n^{\text{core}} = 10, n^{\text{conn}} = 10, n^{\text{dis}} = 0, seed = 100$

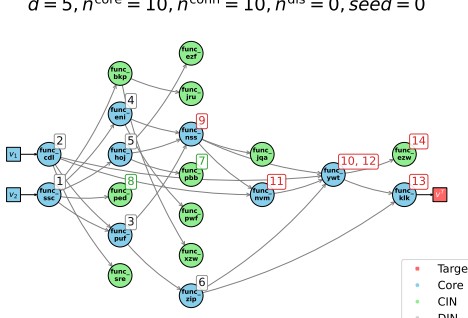 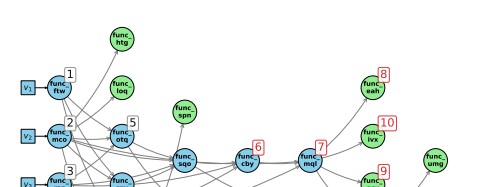

(a) A failure example in which GPT-5 calls irrelevant functions and invoke core functions with wrong input values.

(b) A failure example in which GPT-5 misses required function calls and attempt to call the target function with irrelevant values.

Figure 9: Analysis of GPT-5 function calling errors. The generated graphs feature a depth of 5, comprising 10 core nodes, 10 CINs, and 0 DINs across various random seeds. The number at the top-right of each node denotes the sequential order of function calls made by GPT-5, e.g., "2" indicates the second function called. The color of the number indicates the result: black for correct calls, green for irrelevant nodes (no error), and red for errors.

## 4.7 CASE STUDY OF FUNCTION CALLING FAILURE

To illustrate how LLMs fail in function calling tasks, we present a case study of GPT-5's failure examples in Figure 9. In Figure 9a, GPT-5 incorrectly calls irrelevant functions such as func_pbb and func_ped in the seventh and eighth calls, respectively. Then, we observe that GPT-5 invokes the core function func_nss with wrong input values in the ninth call, i.e., it uses correct variable values from func_puf and func_eni but also uses an incorrect value from func_pbb instead of func_hoj even though it was successfully called in the fifth call. In Figure 9b, GPT-5 misses required function calls such as func_rgy and func_sqo (the blue nodes without a number at the top-right) throughout the entire function calling sequence. As a result, GPT-5 attempts to call the target function func_xrx with irrelevant values from func_qcu and func_otq in the fifth and ninth call. These examples highlight the challenges LLMs face in accurately navigating function call sequences, particularly in avoiding irrelevant functions and ensuring correct input values for core functions. We provide a full function calling sequence of the example in Appendix C.4.

## 5 CONCLUSION

We present FuncBenchGen, a novel framework for generating multi-step function calling benchmarks that address key limitations in existing evaluations. By enabling controllable complexity and contamination-free tasks, FuncBenchGen provides a robust platform for systematically assessing LLMs' function calling capabilities. Our extensive experiments reveal significant performance gaps between reasoning-optimized and general-purpose models. Results also highlight challenges posed by irrelevant functions and long call sequences – even GPT-5 struggles with longer function call sequences. We identify common failure types, and based on this analysis propose a simple yet effective mitigation strategy that significantly improves performance across various models. Overall, FuncBenchGen offers a valuable tool for advancing research into LLM function calling capabilities. Incorporating more complex control flows such as conditional logic and iteration would be our interesting future direction.

## REPRODUCIBILITY STATEMENT

We have made several efforts to ensure the reproducibility of our work. First, the design principles, methodology, and evaluation setup for FuncBenchGen are detailed in Section 3. All models used in the experiments are well-documented in Appendix B.1, with the prompt example in Appendix B.3.

ETHICS STATEMENT

We made use of AI tools such as ChatGPT and Copilot to support coding and refining this paper, but all content was carefully reviewed and edited by us to ensure it adheres to our standards and aligns with our research objectives.

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

# A  ADDITIONAL DETAILS OF GRAPH GENERATION

## A.1  GRAPH GENERATION PROCESS DETAIL

We illustrate the graph generation process in Figure 10. As described in Section 3, the graph generation consists of two main steps: core node creation and irrelevant node addition. In the core node creation step, we first generate a initial graph that is a sequence of nodes based on the specified number of dependency depth as shown in the left in the figure. Then, we iteratively add remaining

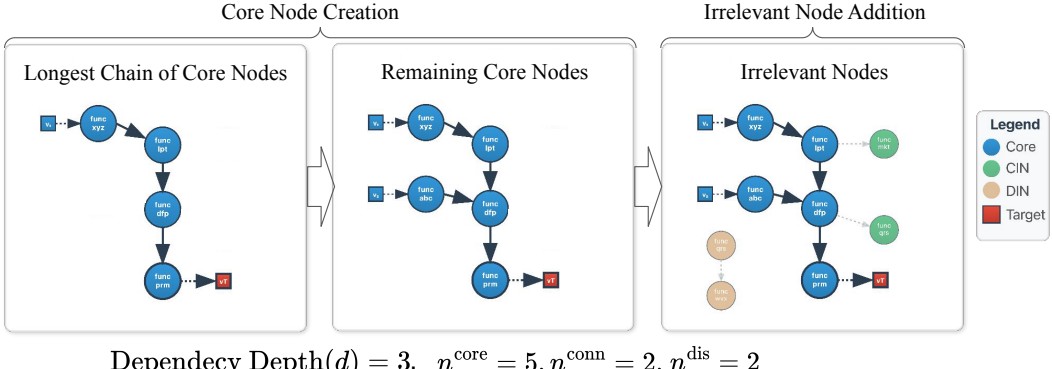

$$\text{Dependecy Depth}(d) = 3, \quad n^{\text{core}} = 5, n^{\text{conn}} = 2, n^{\text{dis}} = 2$$

Figure 10: Graph generation process.

nodes to the graph until reaching the specified number of core nodes while ensuring that the maximum dependency depth is not exceeded (see the middle in the figure). To increase the diversity, we randomly add up to $2n^{\text{core}}$ edges between core nodes while maintaining the acyclic property of the graph. If adding the new node would exceed the maximum dependency depth, we discard that choice and try again. In the irrelevant node addition step, we add irrelevant nodes based on the specified connection type (see the right in the figure). For connected irrelevant nodes, we randomly select parent nodes from the existing core nodes and link the new irrelevant nodes to it as their children. For disconnected irrelevant nodes, we simply add the new irrelevant node without linking it to any core nodes.

## A.2 EXAMPLE GENERATED GRAPHS

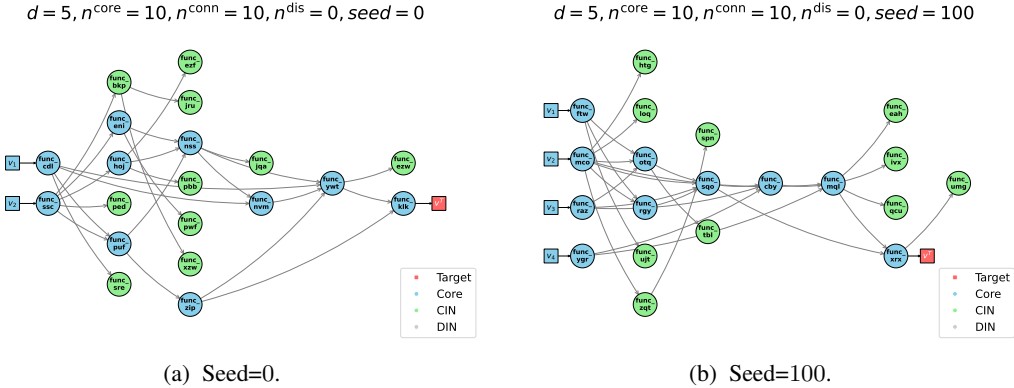

(a) Seed=0.

(b) Seed=100.

Figure 11: Example DAGs with depth 5, 10 core nodes, 10 CINs, 0 DINs, and different random seeds.

To provide concrete examples of our generated graphs, we present figures illustrating DAGs with varying depths and numbers of core and irrelevant nodes used in our experiments. Specifically, Figure 11 displays generated DAGs with a depth of 5, 10 core nodes, 10 CINs, and 0 DINs, generated using different random seeds (0 and 100 in Figures 11a and 11b, respectively). These examples demonstrate how the overall structures of the DAGs can vary significantly—for instance, one graph features two first-layer core nodes while the other has four—while maintaining the same depth and total number of nodes. In our experiments, we use five different random seeds and aggregate the results to ensure robustness against these structural variations.

Next, we provide generated DAGs with depth 3 and 11, 20 core nodes, 5 CINs, and 5 DINs in Figure 12, to demonstrate how the structure changes with varying depths while keeping the number of core and irrelevant nodes constant. Figures 12a and 12b illustrate these graphs, respectively. The depth 3 graph exhibits a wider structure, while the depth 11 graph shows a longer chain of

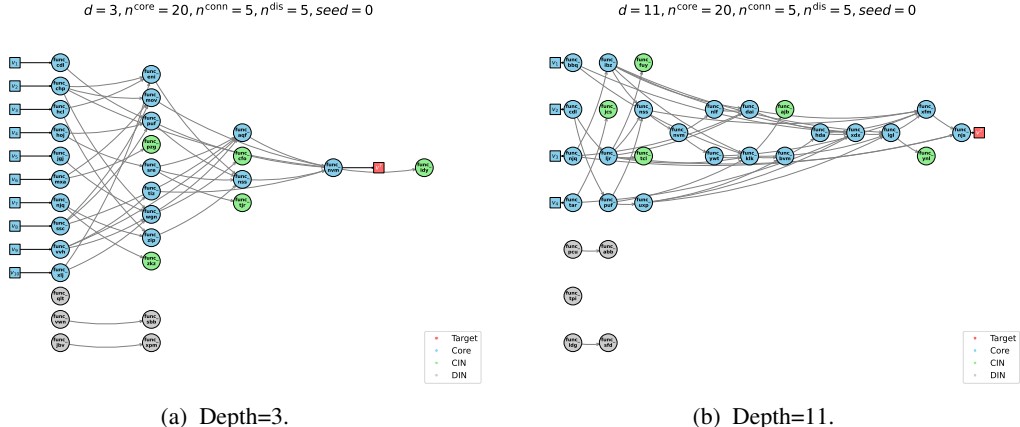

(a) Depth=3.  (b) Depth=11.

Figure 12: Example DAGs with different depths, 20 core nodes, 5 CINs, and 5 DINs.

dependencies. These examples highlight the flexibility of our graph generation process in creating diverse structures that can effectively test the function calling capabilities of LLMs under different complexity levels.

## B ADDITIONAL EXPERIMENT DETAILS

### B.1 MODEL DETAILS

| Model | Size | Context | HuggingFace / API | License |
|---|---|---|---|---|
| GPT-5 (OpenAI, 2025b) | - | 400k | gpt-5-2025-08-07 | OpenAI Service Terms[1] |
| GPT-5-mini (OpenAI, 2025b) | - | 400k | gpt-5-mini-2025-08-07 | OpenAI Service Terms |
| Gemini-2.5-Pro (Comanici et al., 2025) | — | 1M | gemini-2.5-pro | Gemini API Additional Terms of Service[2] |
| Gemini-2.5-Flash (Comanici et al., 2025) | — | 1M | gemini-2.5-flash | Gemini API Additional Terms of Service[2] |
| Qwen-3 (Team, 2025) | 235B | 128k | Qwen/Qwen3-235B-A22B-Instruct-2507 | Apache license 2.0 |
| GPT-4.1 (OpenAI, 2025a) | — | 1M | gpt-4.1-2025-04-14 | OpenAI Service Terms |
| GPT-4.1-mini (OpenAI, 2025a) | — | 1M | gpt-4.1-mini-2025-04-14 | OpenAI Service Terms |

Table 4: Models used in experiments. Model sizes are not publicly disclosed (-).

We summarize the details of the models used in our experiments in Table 4. All models are used via their respective APIs except for Qwen3, which is accessed through HuggingFace. We set the temperature and top-p parameters to $0.0$ and $1.0$, respectively, for all our experiments. For models that do not support the temperature parameter, we use their default settings.

### B.2 TYPES AND SUBTYPES

We observed that if functions were linked based on exact variable names, the problem became too easy. In real-world scenarios, even if a function consumes the output of another function, the names of those variables are rarely the same. Instead these relationships are captured through semantic meaning and data types. For example, if function A outputs a variable of type SQLQuery and function B has an input of type SQLQuery, that often means that function B is designed to consume the output of function A. To reflect this, FuncGenBench links functions based on types, rather than direct variable names. The only exception to this are the input variables listed in the model prompt. Those are linked to function schemas based on direct variable name matching instead.

Additionally, each variable is given both a type and a subtype. A subtype is unique to that variable to allow for unambiguous linking between two functions, but types may be shared across many different variables of completely separate functions. We include both type and subtype to better reflect real-world scenarios where many functions operate on the same kind of data, but only some functions are able to consume the outputs of others in a meaningful way. For example, many functions could accept an argument of type "Employee", but some functions (such as num_managed_employees) would expect that "Employee" to have subtype "Manager."

---

**Function Linking Strategies**

**Linking based on variable names:**

```
Func_yep processes variable mfmjsy to produce variable tcok.

Func_ayj processes variable tcok to produce variable arpl.
```

**Linking based on types:**

```
Func_yep processes variable mfmjsy (type_uxe with subtype_muw) to produce
variable aargww (type_beo with subtype_dej)

Func_ayj processes variable riivq (type_beo with subtype_dej) to produce
variable sjyav (type_wdc with subtype_uqq)
```

---

### B.3 EXAMPLE MODEL INPUT

We provide an example of the model input used in our experiments below.

---

**Example Model Input**

**User Prompt:**
Using the tools at your disposal, use functions until you are able to give me the correct value of variable
bujxye.
Variable mfmjsy = 731
...
You have all the information you need to get the correct result.
**Tool Prompt:**

```
[{'type': 'function',
  'function': {'name': 'func_yep',
   'description': 'Processes variable of (type_uxe with subtype_muw) to
   produce (type_beo with subtype_dej)',
   'strict': True,
   'parameters': {'type': 'object',
    'properties': {'mfmjsy': {'type': 'integer'}},
    'required': ['mfmjsy'],
    'additionalProperties': False}}},
...
]
```

---

## C    ADDITIONAL RESULTS

### C.1    RESULTS OF NO EXTRA SETTING

Table 5 shows the results of all models in the no extra irrelevant node setting. We observe that the
trends of success rates and ACs are consistent with those in Section 4 while the "No Extra" setting
obtains higher success rates in general.

### C.2    RESULTS OF DEPENDENCY DEPTH WITH ERROR BARS

Figure 13 shows the results for each dependency depth with error bars in which the results are
averaged across the numbers of noisy connected nodes.

### C.3    RESULTS OF OTHER DEPENDENCY DEPTHS

Figure 14 shows the results of all models for each dependency depth with 20 core nodes. We
observe similar trends to those in Figure 5. Lower dependency depth leads to higher success rates
across models.

Table 5: Success rates and average function calls (**ACs**) that were actually made by models. Results are aggregated across only no extra irrelevant node configurations and graph dependency depth $\{1, ... n^{\text{core}} - 1\}$, with 5 random trials per configuration. ACs (Succ.) and (Fail.) denote the ACs for successful and failed trials, respectively. "–" indicates that there are no successful trials.

| # core nodes | 5 | | | 10 | | | 20 | | |
|---|---|---|---|---|---|---|---|---|---|
| | Success Rate | ACs (Succ.) | ACs (Fail.) | Success Rate | ACs (Succ.) | ACs (Fail.) | Success Rate | ACs (Succ.) | ACs (Fail.) |
| GPT-5 | 80.0% | 5.0 | 5.0 | 28.9% | 10.4 | 9.6 | 14.0% | 21.0 | 17.6 |
| GPT-5-mini | 20.0% | 5.0 | 4.7 | 8.9% | 10.0 | 8.9 | 4.0% | 20.0 | 17.5 |
| Gemini-2.5-Pro | 65.0% | 5.0 | 5.0 | 20.0% | 10.0 | 12.1 | 10.0% | 20.0 | 24.6 |
| Gemini-2.5-Flash | 35.0% | 5.0 | 0.5 | 22.2% | 10.0 | 0.3 | 10.0% | 20.0 | 0.0 |
| Qwen3 | 10.0% | 5.0 | 5.0 | 4.4% | 10.5 | 9.9 | 4.0% | 21.0 | 18.2 |
| GPT-4.1 | 15.0% | 5.0 | 4.0 | 6.7% | 10.0 | 6.9 | 2.0% | 21.0 | 12.2 |
| GPT-4.1-mini | 10.0% | 5.0 | 4.3 | 0.0% | – | 8.6 | 0.0% | – | 15.0 |

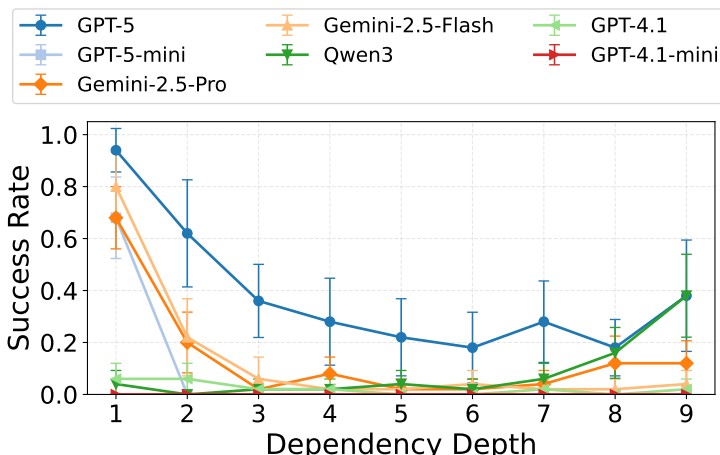

Figure 13: Success rates by the dependency depth with error bars. The number of core nodes is set to 10. Error bars indicate 95% confidence intervals.

Figure 15 shows the results of GPT-5 for each dependency depth with 40 core nodes. Since the core node size is large, the success rates are 0 when dependency depth is greater than or equals 9.

Figure 16 shows the comparison between the baseline and the proposed mitigation strategy with 10 core nodes. We observe similar trends to those in Figure 8.

### C.4 FUNCTION CALLING SEQUENCE EXAMPLE

Figure 17 shows a full function calling sequence example of the case study in Figure 9a.

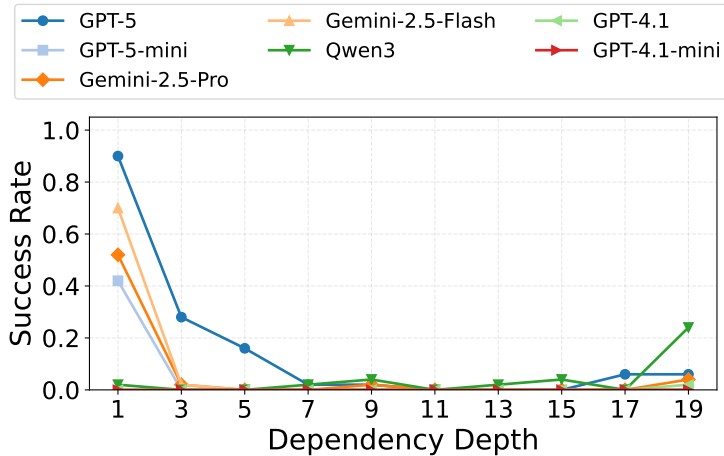

Figure 14: Success rates of all models by the dependency depth. The number of core nodes is set to 20 and the results are aggregated .

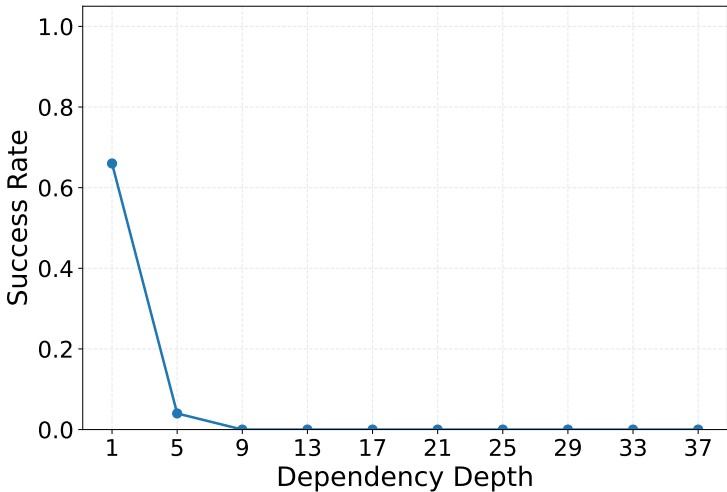

Figure 15: Success rates of GPT-5 by the dependency depth. The number of core nodes is set to 40.

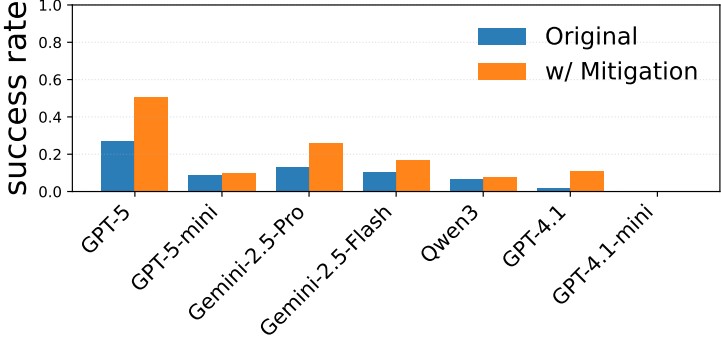

Figure 16: Comparison between the baseline and the proposed mitigation strategy. The number of core nodes is set to 10. The results are averaged across $\{0, 10, 20, 40\}$ CINs, with 5 trials each.

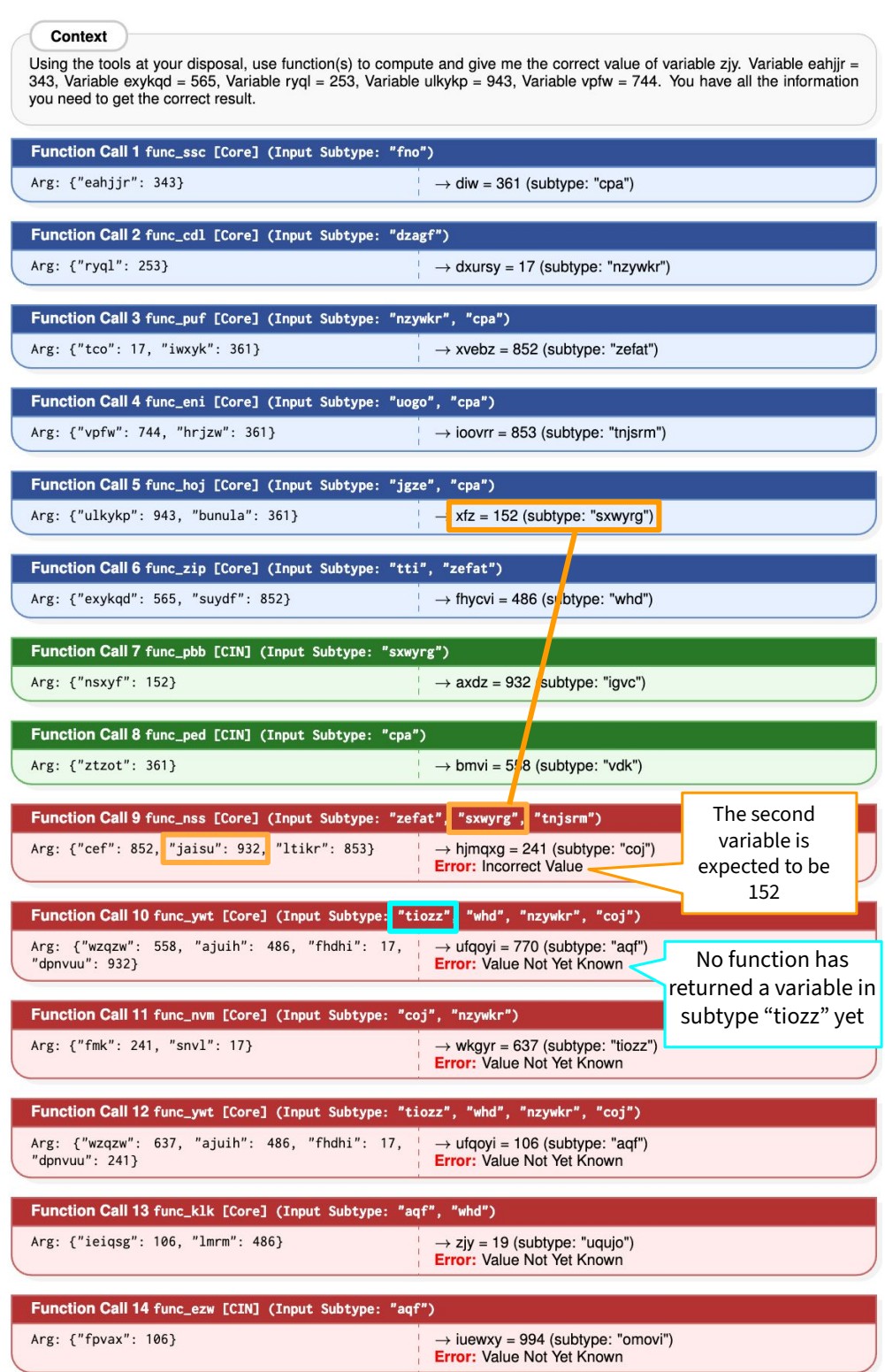

Figure 17: Detailed function calling sequence. The left block shows the input to the function call which the model generates, and the right block shows the output from the function call. Core nodes are highlighted in blue, connected irrelevant nodes (CINs) in green, and failed function calls in red. Errors are annotated below the output values.

