# OpenReview forum: "Towards Reliable Benchmarking: A Contamination Free, Controllable Evaluation Framework for Multi-step LLM Function Calling"
_ICLR.cc/2026/Conference — ICLR 2026 Poster_

### Official Review · Reviewer_Lq7b · 2025-11-01

**Soundness:** 4
**Presentation:** 4
**Contribution:** 4
**Rating:** 8
**Confidence:** 4

**Summary:**

This paper presents FuncBenchGen, a evaluation framework for tool-augmented LLMs. the main contributions are: (1) proposing a framework that can generate contamination-free function-calling tasks. Besides, they conduct extensive experiments with 7 LLMs. They have revealed some interesting findings. Finally, they proposed a simpel augmentation mitigation strategy that restates variable values from prior calls.

**Strengths:**

1. Overall, the presentation is good. The paper is well-written and easy to follow.
2. The contribution is significant, and many experiments are conducted for demonstration. They proposed a evaluation framework that is contamination-free, requiring many task complexities including function, depth and irrelevant function types. Their experiments with 7 LLMs, including the most-up-dated GPT5.

**Weaknesses:**

1. failure analysis and mitigation strategy are not extensively studied. For example, how do you find these 4 types of these failures? how it widely spreading in the cases? How do you define the boundary of each failure type? Section 4.6.1 mitigation strategy, is there more experiments to demonstrate its effectiveness?

**Questions:**

See above.

---

> ### Author Response · Authors · 2025-11-25
> **Author Response**
>
> Thank you for recognizing the value of our function calling benchmark and for your constructive feedback. Below are our responses to your points:
>
> - **W1:** Request for more details of failure analysis.
>   We address this concern in our general response (please refer to “Failure Types” in G4). In short, we clarified how we define the four failure types and provided a case study with concrete failure examples.
> - **W2:** More experiments of mitigation strategy
>   We additionally conducted experiments with 10 core nodes, maintaining all other settings as described in Figure 8\. The results are shown in our general response (please refer to “Additional Results of Mitigation Strategy” in G4). The results demonstrate the same trend that our mitigation strategy consistently improves the performance in multi-step function calling tasks. We have included the results in Figure 16 (Appendix C.3) of the revised paper.
>
>
> We appreciate your feedback again. We hope our additional results are interesting and address your concerns.

---

### Official Review · Reviewer_cs9L · 2025-11-01

**Soundness:** 3
**Presentation:** 3
**Contribution:** 3
**Rating:** 6
**Confidence:** 3

**Summary:**

This paper presents FuncBenchGen, an automated contamination-free benchmark for testing tool-augmented LMs on multi-step function calls with fine-grained control over difficulty. It treats tool use as walking a hidden function-dependency DAG and generates synthetic function schemas at eval time to prevent train/test leakage. The setup lets you tune depth, function count/types, and add distractors. The authors test seven leading LMs, analyze where they fail, and show a simple fix—explicitly restating variables—significantly boosts success.

**Strengths:**

- The framework generates tasks and functions on the fly, preventing pretraining/test-time leakage and removing the data-contamination issues that plagued prior function-calling benchmarks.
- FuncBenchGen exposes fine control over function-set size, dependency depth (DAGs), and irrelevant distractors (connected/disconnected), enabling targeted evaluation along key axes (Table 1).
- The study is broad and careful. The leading LMs are tested across graph sizes, complexities, and distractors; The results (Table 2; Fig. 3–4) give a multi-view of behavior, and the executor breakdown (Table 3) shows variable-tracking errors dominate—pointing to clear fixes.
- The proposed fix is simple and effective. Based on the reported results, for example, GPT-5 show gains from 62.5% to 81.3%. Besides, modeling multi-step tool use as DAGs and enforcing schema/type matching aligns with real API workflows and applications.

**Weaknesses:**

- The missing prior work and limited experiments. The study does not directly discuss or compare to the prior work on contamination-free or fine-grained function-calling benchmarks (HammerBench [1], LiveBench [2], etc.). This may significantly limits novelty and contributions. This paper should include the missing work also in Section 2, so to more competently introduce the related work and explain how FuncBenchGen is distinct in controllability, expressivity, or realism.
- Shall include more examples to better show the motivations and potentials/limitations of FuncBenchGen. For example, the paper may show qualitative examples of actual failed call traces or state-tracking failures, such as incorrect call propagation, classic state-tracking errors. Besides, it shall be better if this paper shows more complex control flow, such as conditional logic, iteration, probabilistic outputs, which is commonly appearing in real-world systems.
[1] Wang, Jun, et al. "Hammerbench: Fine-grained function-calling evaluation in real mobile device scenarios." arXiv preprint arXiv:2412.16516 (2024).
[2] White, Colin, et al. "LiveBench: A challenging, contamination-limited LLM benchmark." arXiv preprint arXiv:2406.19314 (2024).

**Questions:**

- Could you share more about how FuncBenchGen samples DAGs? In particular, how do you make sure the sampled graphs are truly representative for a given size and depth?
- How well does the synthetic type-matching approach hold up with real-world function calls where types are deeply nested and often ambiguous? What is the suggested way to use real API sets to stress-test realism and semantic complexity?

---

> ### Author Response · Authors · 2025-11-25
> **Author Response**
>
> Thank you for your constructive feedback and questions. Below are our responses to each of your points:
>
> - **W1:** Suggestion to compare with prior benchmarks.
>   To address this concern, we expanded our discussion of related work and the updated version has been uploaded, in which all edits are highlighted in blue. Please refer to G3 in our general response for more details.
> - **W2:** Request for more examples (e.g., failures, complex flows).
>   Thank you for this suggestion. We additionally included qualitative examples of actual failed call traces in Figure 9 in Section 4.7 of the revised paper (please refer to "Case Study of Function Calling Failure" in G4). We also discuss the possibility of incorporating more complex control flows, such as conditional logic and iteration as our future work, in the conclusion to better reflect real-world use cases.
> - **Q1:** Request for details on the DAG sampling method.
>   We address this concern in our general response (please refer to G2). In short, we provide actual generated graph examples and a figure that explains the graph generation process step-by-step in the revised paper.
> - **Q2:** Question on applying synthetic types to real-world scenarios.
>   We appreciate this important question. While our current benchmark focuses on synthetic type-matching to ensure controllability and systematic evaluation, we acknowledge that real-world function calls often involve deeply nested and ambiguous types. To better stress-test realism and semantic complexity, we suggest using real API sets with rich type hierarchies and incorporating semantic annotations that capture the intended use of each function. This could involve leveraging existing API documentation and usage examples to create more semantically meaningful tasks. We will explore this direction in future work.
>
> Thank you again for your constructive feedback. We hope our responses and the revised manuscript address your concerns.

---

### Official Review · Reviewer_Hw9C · 2025-11-06

**Soundness:** 3
**Presentation:** 3
**Contribution:** 2
**Rating:** 4
**Confidence:** 4

**Summary:**

The paper introduces FuncBenchGen, a benchmark for tool-augmented language models. The authors synthetically generate function-calling tasks in the form of DAG's, with each node taking in some input/output variables and edges representing function composition. The authors benchmark seven LLMs on FuncBenchGen and propose a mitigation strategy to improve performance on their benchmark.

**Strengths:**

- The proposed benchmark is synthetic, which allows the authors to study various properties in a controlled manner. The authors vary multiple dimensions such as the number of irrelevant premises, the size of the graph, and so on.
- An aspect of tool-use is measured: being able to retrieve the correct functions and use them in a way that allows the LLM to solve the execution prediction problem.
- The paper is presented clearly and the evaluation is sound. Most SoTA models are tested on the benchmark, and quite a few of them struggle to complete the task.

**Weaknesses:**

- While this synthetic task is interesting from an analysis point of view, it is unclear how it applies to real-world function calling scenarios. While the authors briefly address this in Sec 3.3, I find the abstraction to simplify the task a lot and would like to see at least one of the described real-world scenarios evaluated in a more realistic setting to validate the simplified formulation being a reliable proxy.
- The papers fails to mention a large relevant body of literature focused on following the flow of programs and logic chains (e.g. https://arxiv.org/abs/2209.00840, https://arxiv.org/abs/2401.03065, https://arxiv.org/abs/2403.16437). These tasks, in spirit, are very similar, as they relate to being able to follow relevant chains of logic in a graph. Many of the research questions have been studied in similar contexts (e.g. RQ2 in https://arxiv.org/abs/2302.00093, RQ3 in https://arxiv.org/abs/2406.17169).

**Questions:**

- How would fine-tuning on the task affect the performance on the benchmark? One advantage of the synthetic setup is that it is very easy to generate more examples of the task to study something like this.
- While you mention is an abstraction of a more complex real-world task, do you have any evidence performance on this task correlates in some way to performance on a real-world function calling task? Often, tasks that are equivalent down to some abstraction can have large differences when evaluating a model on them.

---

> ### Author Response · Authors · 2025-11-25
> **Author Response**
>
> Thank you for your valuable feedback and questions. Below are our responses to each of your points:
>
> - **W1 & Q2:** Request to clarify real-world applicability and the correlation to real-world performance.
>   Please refer to G1 in our general response. In short, we provide further details connecting our setting with real-world use cases and show the existing benchmarks with real-world tools and APIs report comparable performance to our synthetic benchmark, indicating that long-term interactions with real-world tools remain challenging, which is consistent with our findings in synthetic function-calling tasks. This similarity suggests that our findings generalize to real-world function-calling settings.
>
> - **W2:** Suggestion to include related literature.
>   To address this concern, we expanded our discussion of related work and the updated version has been uploaded, in which all edits are highlighted in blue. Please refer to G3 for more details.
>
> - **Q1:** Question about the potential effect of fine-tuning.
>   Thank you for suggesting this interesting direction. We agree that fine-tuning could potentially improve performance, especially given the synthetic nature of the task which allows for generating large amounts of training data. A potential direction would be to generate text reasoning paths that encourage models to better think long dependency of function calls, instead of simply learning right answers. We will explore this in future work.
>
> Thank you for your detailed review. We hope our responses and the revised manuscript address your concerns. If you take these points into consideration and find more value in our study, we would be grateful if you could adjust your score in a positive way.

---

### Official Review · Reviewer_UX5z · 2025-11-06

**Soundness:** 2
**Presentation:** 2
**Contribution:** 2
**Rating:** 4
**Confidence:** 4

**Summary:**

This paper proposes FuncBenchGen, a contamination-free evaluation framework designed to address the key limitations in existing benchmarks for multi-step function calling of TaLMs. Current benchmarks face the risk of data contamination and lack fine-grained control over task complexity factors such as function dependency depth, the number of core functions, and the presence of irrelevant functions. FuncBenchGen resolves these issues by formulating multi-step function calling as traversal over a hidden function-dependency DAG. The framework dynamically generates synthetic functions and tasks during evaluation to ensure no data leakage, while allowing users to precisely adjust complexity parameters, including the number of core functions, dependency depth, and the type and quantity of irrelevant distractor functions. Extensive experiments on seven state-of-the-art large language models show that reasoning-optimized models consistently outperform general-purpose models, with GPT-5 achieving the highest success rate but performing poorly in long-sequence tasks. CINs severely impair performance by interfering with relevant paths, and model performance declines sharply as dependency depth increases. The most common failure mode is the use of "unknown variable values," while syntax errors are relatively rare. A simple mitigation strategy, explicitly restating all known variable values at each step, can significantly improve performance.

**Strengths:**

1.Proposes an innovative DAG-based FuncBenchGen framework enabling fine-grained complexity adjustment.
2.Conducts extensive experiments on 7 state-of-the-art LLMs with rigorous variable control and repeats several times, yielding reliable results.
3.Presents a practical lightweight mitigation strategy with significant performance gains without model modification/training.

**Weaknesses:**

1.Given a task, how can a framework automatically generate a test suite with a specified depth of n? How should the functionality of each core node be constructed? It is crucial to ensure that exactly n nodes are sufficient and necessary to complete the task and that each node is feasible; however, this is not mentioned in the text.
2.There is a discrepancy between the "type + subtype" matching of synthetic functions and the parameter semantic matching of real APIs. The framework's migratability in real API scenarios has not been verified.

**Questions:**

1.After a task is split into m core nodes, will m function APIs actually be built as tools for the large model to call? If querying addresses, web searches, etc., are involved, how is each of these tools constructed?
2.When the model is only provided with the meta-information (function description, type + subtype annotations) of synthetic functions without knowing their preset "input→output" mapping rules, how does the model ultimately obtain the expected target variable value? How does the model infer the correct function call sequence without knowning the semantic meaning? If the aforementioned questions and shortcomings are explained, a reassessment of the score may be conducted.

---

> ### Author Response · Authors · 2025-11-25
> **Author Response**
>
> Thank you for your thoughtful feedback and questions. Below are our responses to each of your points:
>
> - **W1:** Request for details on test suite generation (depth n).
>   To address this concern, we provide additional details on our graph generation process and added figures that explain the graph generation process step-by-step (please refer to "Additional Graph Generation Details" in G2 of our general response). In short, we ensure that the generated graphs satisfy the specified depth and number of core nodes by first creating a base structure that meets the depth requirement and then iteratively adding nodes while checking for depth constraints.
> - **W2 & Q2:** Questions on bridging synthetic types and real API semantics and how models infer sequences without semantics.
>   We discuss this aspect of the paper in more depth in G1 in our general response. In short, our type/subtype annotations serve as a proxy for the semantic relationships that implicitly map the inputs and outputs of functions together in real use cases.
>   In addition, we observe that the models are able to find the correct sequence of function calls because \- recognizing that the functions are linked via type/subtype \- they are able to plan ahead and recursively identify which functions produce the required inputs to reach the desired variable. We infer the presence of implicit planning from the accuracy and efficiency of the generated call sequences in our experiments, which show that the models rarely make many more function calls than required. They are not performing a breadth-first search, calling every function that has the inputs for, but are instead implicitly reconstructing and navigating the original graph structure based on the linking of types and subtypes.
> - **Q1:** Question on the construction of function API tools.
>   As you pointed out, the framework generates m distinct function APIs corresponding to the core nodes, but these are constructed as synthetic, deterministic schemas rather than functional real-world software. Instead of implementing actual utilities for operations like address queries or web searches, FuncBenchGen abstracts these real-world scenarios into a dependency graph where tools are linked solely by semantic type and subtype matching; consequently, executing a complex task is simulated by passing integer values between functions to verify the model's logical planning, rather than interacting with a live external environment which is sometimes uncontrollable.
>   In the revised paper, we additionally provide Figure 17 in Appendix, which explains how an actual function calling sequence looks like. We hope this gives better understanding.
>
>
> We would appreciate it if you could conduct a reassessment of the score after reviewing our responses and the revised manuscript.

---

### Official Review · Reviewer_ncD9 · 2025-11-10

**Soundness:** 3
**Presentation:** 2
**Contribution:** 3
**Rating:** 4
**Confidence:** 5

**Summary:**

This paper proposes FuncBenchGen, a benchmark framework for evaluating LLMs on multi-step function-calling tasks under controlled and contamination-free conditions. The benchmark allows fine-grained control over task complexity (e.g., function count, depth, and distractors) and reveals key limitations in models' state tracking. A simple mitigation strategy, restating known variables, significantly improves performance.

**Strengths:**

1. This paper proposes a novel benchmark framework that addresses a critical and under-explored challenge, evaluating LLMs on multi-step tool use in a contamination-free, controlled setting.
2. Experiments are comprehensive, covering a range of models and various settings.
3. The proposed mitigation strategy, while simple, is effective and general.

**Weaknesses:**

1. The paper suffers from significant formatting and presentation problems. For example, the abstract is split into three paragraphs, disrupting its coherence.
2. The paper does not provide concrete examples, making it difficult to understand what the actual data instances look like. Without such examples, it is unclear how complex the generated tasks truly are, what kind of reasoning is required at each step, or how function inputs and outputs are represented.
3. Function schemas and variable types are generated entirely at random without semantic constraints, which reduces interpretability and realism. As a result, the benchmark evaluates abstract structural reasoning rather than meaningful function-calling behavior, limiting external validity.
4. Tasks are purely synthetic, with limited connection to real-world API use or natural semantics.
5. The proposed "variable reminder" mitigation, while effective, is arguably a hack rather than a principled solution. It sidesteps the problem rather than enabling models to learn better internal state tracking.

**Questions:**

1. Since function schemas and variable types are randomly assigned, how do you ensure that the generated tasks remain semantically coherent or representative of real-world function-calling scenarios?
2. How do you validate that randomly generated functions produce diverse yet meaningful reasoning chains, rather than trivial or redundant ones?
3. The process for generating CIN and DIN nodes appears partly heuristic. Could you clarify how these nodes are chosen and connected in practice?

---

> ### Author Response · Authors · 2025-11-25
> **Author Response**
>
> We appreciate your constructive feedback and questions. Below are our responses to each of your points:
>
> - **W1:** Feedback on formatting and presentation.
>   Thank you for pointing out the formatting issues. We carefully revised the presentation to enhance conciseness and coherence. Concretely, we reduced the abstract by 90 words (from 316 to 226), condensing it into a single paragraph. While we have done our best to improve the paper organization, formatting, and presentation, we would greatly appreciate it if you could share any specific sections you found unclear. This would help us ensure that we address all your concerns effectively.
>
> - **W2**: Request for more concrete examples.
>   To address this concern, we additionally provide concrete generated graph examples in the revised paper and discuss the details in our general response (please refer to G2). We also include an additional figure that explains how an actual function calling process looks like in Figure 17 in Appendix of the revised paper.
>
> - **W3**, **W4 & Q1:** Clarification on the realism of generated schemas, the connection to real-world tasks, and semantic coherence.
>   We address this concern in our general response (G1). Briefly, while our focus is on structural reasoning rather than semantic coherence, the difficulty observed in our synthetic function-calling tasks, which are long-term interaction challenges, is consistent with findings on existing benchmarks that use real-world tools and APIs, which report comparable performance on Gemini-2.5-Pro. This similarity suggests that our findings generalize to real-world function-calling settings.
>
> - **W5:** Feedback on the proposed mitigation strategy.
>   We address this concern in our general response (please refer to "Mitigation Strategy" in G4). In short, a simple reminder technique has also been shown to be effective in other tasks. While we acknowledge that this approach is simple, our experiments demonstrate its effectiveness in improving performance on multi-step function-calling tasks.
>
> - **Q2:** Question on validating the meaningfulness of reasoning chains.
>   To address this concern, we provide additional details on our graph generation process and added figures that explain the graph generation process step-by-step (please refer to "Additional Graph Generation Details" in G2). In short, by controlling the dependency depth and the number of core nodes, we ensure that the generated tasks require non-trivial reasoning chains to reach the target variable.
>
> - **Q3:** Request for details on CIN/DIN node generation.
>   While our graph generation process involves some heuristics, our focus is on constructing controllable and diverse function-calling tasks. We ensure that CIN and DIN nodes are generated in a way that maintains the overall structure and complexity of the task while reflecting the presence of irrelevant information which often happens in real-world function calling tasks. Please refer to G2 for more details on our graph generation process.
>
>
> Thank you again for your constructive feedback. We hope our responses and the revised manuscript address your concerns. If you take these points into consideration and find more value in our study, we would be grateful if you could adjust your score in a positive way.

---

### Author Response · Authors · 2025-11-25
**General Response 1**

We are grateful for the valuable feedback provided by all reviewers on our paper. The thoughtful suggestions and constructive comments not only helped us improve the overall quality of the work but also strengthened the clarity and validity of our findings and claims. In response, we have incorporated the following changes in the revised version of the paper:

## G1. Synthetic Nature and Lack of Real-World Validity

Thank you for raising important concerns regarding the synthetic nature of our benchmark and its applicability to real-world function calling scenarios. We believe that our synthetic framework is a valid proxy for real-world function-calling ability for several reasons. As discussed in "Real-world Applicability" in Section 3.3, the presented graph structure naturally follows the same patterns as the dependencies of real function-calling scenarios, and in many cases it is easy to map from one to the other. To further illustrate this, we provide an additional figure (Figure 3 in Section 3.3 of the revised paper) that demonstrates how a real-world function calling task can be abstracted into our synthetic benchmark format. In addition, our implementation still relies on the built-in function-calling API of these models \- the models aren’t simply navigating a graph that represents function calls, but are actually calling functions and getting responses back in the standard way. The difference between FuncBenchGen and other tool-calling benchmarks is that FuncBenchGen’s functions lack semantic meaning. In other words, the synthetic functions don’t represent concrete tasks (such as executing SQL queries) but rather abstract and arbitrary tasks (transforming data of type ABC into type XYZ), which is precisely what allows us to rigorously control the task difficulty and conduct comprehensive experiments. To mitigate this difference in semantic meaning, we designed the type/sub-type mapping system to mimic real-world function semantics as much as possible and bridge the gap between synthetic and real-world function sets.

For example, in a real API use-case, many functions may take arguments of type “string” but one expects a table name as input, another expects a SQL query, and so on. Each of these “subtypes” are strings, but represent different kinds of data and are often unique to each method. Subtypes, in our implementation, represent this separation between different kinds of data that still share the same data type. We never explicitly tell the model any mapping rules that imply that functions are linked via types and subtypes.  We provide only the initial variable values with their types/subtypes, function schemas, and the target variable with its type/subtype. Models must infer that functions can be chained when output types match input types \- the same reasoning required in real-world API usage. Though our functions still lack inherent semantic meaning, our explicit type/subtype annotations serve as a proxy for the semantic relationships that guide real function composition.

As for the function calling performance, we compare our results with existing benchmarks (ToolHaystack \[1\] and PaperArena \[2\]) that utilize real-world tools and APIs to demonstrate the relevance of our findings. The ToolHaystack paper discusses that the performance of GPT-4o drops from 83.33% to 35.87% when they compare BFCL (Berkeley Function Calling Leaderboard \[3\], which has 4.2 average turns) with ToolHaystack (32 average turns). Also, PaperArena reports that the average accuracy of the state-of-the-art agent (Gemini-2.5-Pro) is only 38.78%, which drastically drops to 18.47% on the hard subset where reasoning chains average approximately 10 steps and can exceed 30 steps in complex scenarios. These numbers are comparable to our results in Table 2, where Gemini-2.5-Pro achieves 46.5% for 5 core nodes and 14.4% for 10 core nodes.

The parallel trends in performance across both settings suggest that our synthetic DAG-based framework captures the same underlying reasoning difficulties present in real-world tool-use tasks, providing evidence that our evaluation setting is both realistic and generalizable. Moreover, the results indicate that long-term interactions with real-world tools remain challenging, which is consistent with our findings in synthetic function-calling tasks. However, these benchmarks do not provide systematic control over task complexity, such as dependency depth and the number of core nodes. In contrast, our benchmark allows for precise manipulation of these factors, enabling a more granular analysis of model capabilities.

---

> ### Author Response · Authors · 2025-11-25
> **General Response 2**
>
> \[1\] Kwak, Beong-woo, et al. "ToolHaystack: Stress-Testing Tool-Augmented Language Models in Realistic Long-Term Interactions." EMNLP findings (2025).
> \[2\] Wang, Daoyu, et al. "PaperArena: An Evaluation Benchmark for Tool-Augmented Agentic Reasoning on Scientific Literature." arXiv preprint (2025).
> \[3\] Patil, Shishir G., et al. "Gorilla: Large language model connected with massive apis." NeurIPS (2024).
>
> ## G2. Examples, and Generation Details
>
> We appreciate the reviewers highlighting the need for concrete examples and more details regarding our graph generation process. We will address each of these points.
>
> ### Examples of Generated Graphs
>
> To provide concrete examples of our generated graphs, we have added new figures in Appendix A, which illustrate generated DAGs with varying depths and numbers of core and irrelevant nodes, which were actually used in our experiments. These figures are included in the revised version of the paper. Specifically, we first present generated DAGs with depth 5, 10 core nodes, 10 CINs, 0 DINs, and different random seeds (0 and 100), where we show the overall structures of the DAGs are different, e.g., one has two first layer core nodes and the other has four first layer nodes, while maintaining the same depth and number of nodes. Note that we use five different random seeds and aggregate the results in our experiments to ensure robustness.
> Next, we provide generated DAGs with depth 3 and 11, 20 core nodes, 5 CINs, and 5 DINs, to demonstrate how the structure changes with varying depths while keeping the number of core and irrelevant nodes constant. The depth 3 graph exhibits a wider structure, while the depth 11 graph shows a longer chain of dependencies. These examples highlight the flexibility of our graph generation process in creating diverse structures that can effectively test the function calling capabilities of LLMs under different complexity levels.
>
> ### Additional Graph Generation Details
>
> To show further graph generation details, we additionally provide a figure that explains how FuncBenchGen samples DAGs with specific depth and number of core nodes. The figure is added in the Appendix A (Figure 10).
> In the figure, we illustrate the step-by-step process of sampling a DAG with depth 3, 5 core nodes, and 4 irrelevant nodes. We start by creating an initial graph that is a sequence of nodes to ensure the required depth. Then, we iteratively insert remaining nodes to the graph by randomly adding new parent nodes to the existing core nodes until reaching the specified number of core nodes. If adding the new node would exceed the maximum dependency depth, we discard that choice and try again. In the irrelevant node addition step, we add irrelevant nodes based on the specified connection type. For connected irrelevant nodes (CIN), we randomly select parent nodes from the existing core nodes and link the new irrelevant nodes to it as their children. For disconnected irrelevant nodes (DIN), we simply add the new irrelevant node without linking it to any core nodes.
> Through this process, our graph generation ensures that generated graphs satisfy all input conditions such as the dependency depth and the numbers of core nodes, CINs, and DINs.
>
> We appreciate the reviewers’ suggestion again. By addressing it, the descriptions about our graph generation process have become more clear and reader-friendly.
>
> ## G3. Additional Prior Work
>
> We expanded our discussion of related work and the updated version has been uploaded, in which all edits are highlighted in blue. First, we described that FOLIO \[1\] and Multi-LogiEval \[2\] are examples of complex reasoning benchmarks using first order logic, while CRUXEval \[3\] and REval \[4\] target code generation and program reasoning. These benchmarks focus on code generation/reasoning rather than function calling that we focus on, where external tools are treated as black-box APIs with input–output specifications and natural-language descriptions.
>
> Also, we include LiveBench \[5\], which aims to mitigate data contamination but does not consider function calling. HammerBench \[6\] evaluates function calling with real functions, yet it does not provide controllable task complexity such as dependency depth and the number of core nodes.
> We additionally discuss existing work \[7\] demonstrating that distracted information hurts the performance of models in our experiments. While this trend is observed in our connected irrelevant node (CIN) setting, our experiments also show that disconnected irrelevant nodes sometimes boost performance, which was not discussed in the previous work.
>
>
> \[1\] Han, Simeng, et al. "Folio: Natural language reasoning with first-order logic." EMNLP (2024).
> \[2\] Patel, Nisarg, et al. "Multi-LogiEval: Towards Evaluating Multi-Step Logical Reasoning Ability of Large Language Models." EMNLP (2024).

---

> ### Author Response · Authors · 2025-11-25
> **General Response 3**
>
> \[3\] Gu, Alex, et al. "CRUXEval: A Benchmark for Code Reasoning, Understanding and Execution." ICML PMLR, (2024).
> \[4\] Chen, Junkai, et al. "Reasoning Runtime Behavior of a Program with LLM: How Far Are We?." ICSE, IEEE Computer Society, (2024).
> \[5\] White, Colin, et al. "LiveBench: A Challenging, Contamination-Limited LLM Benchmark." ICLR (2025).
> \[6\] Wang, Jun, et al. "Hammerbench: Fine-grained function-calling evaluation in real mobile assistant scenarios." ACL findings (2025).
> \[7\] Shi, Freda, et al. "Large language models can be easily distracted by irrelevant context." ICML PMLR, (2023).
>
> ## G4. Scope of Failure Analysis and Mitigation
>
> We appreciate the reviewers' feedback regarding the depth of our failure analysis and mitigation strategies. All the raised concerns have been carefully considered, and we have revised and uploaded the updated manuscript accordingly. Below are the specific changes made to address these concerns:
>
> ### Failure Types
>
> Our current analysis focuses on four primary failure types that cover all systematically detectable errors, as they correspond to the hierarchical prerequisites required for any valid function execution: 1\) function existence errors, 2\) argument mismatch errors, 3\) incorrect call sequence errors, and 4\) state-tracking errors. These failure types can be considered in sequence (1–4) using if-branches, and any error arising from a function call will fall into one of these categories. This content has been added at the beginning of Section 4.6.
>
> ### Case Study of Function Calling Failure
>
> To provide examples that demonstrate how LLMs fail in function calling tasks, we additionally presented a case study of GPT-5's failure examples in Figure 9 in Section 4.7. Specifically, we show two examples where GPT-5 struggles to correctly call functions due to incorrect call sequences and state-tracking errors, which are the major failure types observed in our experiments.
> In the first example, while GPT-5 calls functions in the correct order, it mistakenly uses an irrelevant variable as an argument for a function call, leading to an "Incorrect Value" error. In the second example, GPT-5 fails to call the necessary intermediate functions in the correct order and then attempts to call functions that depend on the outputs of those missed functions, leading to "Value Not Yet Known" errors.
> To give more detailed function calling results, we also additionally included the full function calling sequence and the tool responses in Figure 17 in Appendix.
>
> ### Mitigation Strategy
>
> Our proposed "variable reminder" mitigation strategy is designed to directly address state-tracking errors which are also observed in other tasks that require long-term dependency tracking \[1\]. While we acknowledge that this approach is simple and similar to techniques used in prior work, our experiments demonstrate its effectiveness in improving performance on multi-step function-calling tasks. We view this as an initial step towards more sophisticated mitigation strategies and plan to explore additional methods in future work.
>
> \[1\] Tan, Zhen, et al. "In prospect and retrospect: Reflective memory management for long-term personalized dialogue agents." ACL (2025).
>
> ### Additional Results of Mitigation
>
> To further validate the effectiveness of our mitigation strategy, we have conducted additional experiments with 10 core nodes, maintaining all other settings as described for Figure 8, with the exception of the number of core nodes. Table X demonstrates the results, showing the same trend that the mitigation consistently improves performance across all evaluated models. We have added this result in Figure 16 (Appendix C.3) of the revised paper.
>
> Table X: Comparison  between the baseline and the proposed mitigation strategy. The number of core nodes is set to 10\. The results are averaged across {0, 10, 20, 40} CINs, with 5 trials each.
>
> | Model | Original | w/ Mitigation |
> | :---- | :---- | :---- |
> | GPT-5 | 27.22% | 50.56% |
> | GPT-5-mini | 8.89% | 10.00% |
> | Gemini-2.5-Pro | 13.33% | 26.11% |
> | Gemini-2.5-Flash | 10.56% | 16.67% |
> | Qwen3 | 6.67% | 7.78% |
> | GPT-4.1 | 1.67% | 11.11% |
> | GPT-4.1-mini | 0.00% | 0.00% |

---

### Meta-Review · Area_Chair_WTKe · 2025-12-29

**Summary:**

The paper introduces FuncBenchGen, a framework designed to evaluate tool-augmented language models in a contamination-free and controllable manner. By modeling multi-step function calling as a traversal over a hidden directed acyclic graph (DAG), the framework generates synthetic function schemas and tasks on the fly. This approach allows for precise manipulation of task complexity, such as dependency depth and the number of irrelevant "distractor" functions. The authors benchmark seven state-of-the-art LLMs, revealing that reasoning-optimized models outperform general-purpose ones, though performance degrades significantly with increased graph depth. Furthermore, the study identifies state-tracking as a primary failure mode and proposes a simple yet effective mitigation strategy—explicitly restating variable values—which yields substantial performance improvements without model retraining.

**Reviewer Concerns:**

1. The most significant concern, shared by Reviewers ncD9, UX5z, and Hw9C, questioned the external validity of the benchmark. They argued that because the function schemas and types are randomly generated without inherent semantic meaning, the tasks might evaluate abstract structural reasoning rather than realistic API usage. Reviewers questioned whether performance on these synthetic DAGs correlates with real-world scenarios where parameter semantics are complex and ambiguous.

The authors provided a compelling rebuttal by cross-validating their results against existing benchmarks that use real-world tools, such as ToolHaystack and PaperArena. They demonstrated that the performance degradation trends observed in SOTA models (like Gemini-2.5-Pro) on these real-world benchmarks closely mirror the results obtained in FuncBenchGen. The authors argued that the "type/subtype" matching mechanism serves as an effective proxy for semantic relationship planning. The authors have effectively addressed the core of this concern. While the benchmark remains synthetic, the empirical evidence showing a strong correlation with real-world difficulty suggests that FuncBenchGen effectively isolates the *reasoning* component of function calling. While the reviewers who prioritized "realism" may still harbor philosophical reservations about synthetic data, the demonstrated utility of the framework for controllable stress-testing is a strong counter-argument.

2. Several reviewers (ncD9, UX5z, cs9L) noted that the initial submission lacked concrete examples, making it difficult to visualize the generated tasks, the graph generation process, or the specific nature of the failures. Reviewer ncD9 also highlighted formatting issues, while Reviewer UX5z asked for technical details on how the DAGs are sampled to ensure depth constraints.

The authors responded by significantly overhauling the presentation. They added comprehensive figures in the appendix illustrating generated DAGs of varying depths, step-by-step diagrams of the graph generation algorithm, and qualitative examples of function call traces and failures. They also corrected the formatting issues. These revisions fully resolve the presentation-related concerns. The technical clarity regarding how the benchmark ensures "sufficiency and necessity" of nodes in the generated graphs is now sufficient for reproducibility.

3. Reviewer Lq7b asked for a deeper definition of failure types and more evidence regarding the mitigation strategy, while Reviewer ncD9 initially viewed the "variable reminder" mitigation as a "hack" rather than a fundamental solution. Reviewer cs9L requested actual trace examples of failures.

In response, the authors clarified the hierarchical definition of their four failure types (existence, argument mismatch, sequence, and state-tracking) and provided specific case studies of model failures. Crucially, they conducted additional experiments with the mitigation strategy on tasks with 10 core nodes, showing consistent performance gains across all models (e.g., doubling GPT-5's success rate). Characterizing the mitigation as a "hack" overlooks its value as an empirical finding; it highlights a specific weakness in current LLM context management. The additional data solidifies the claim that this simple prompting strategy is a valuable insight for practitioners.

4. Reviewers Hw9C and cs9L noted missing references to relevant literature, specifically regarding logic chain reasoning (e.g., FOLIO) and other recent benchmarks like HammerBench and LiveBench.

The authors have updated the manuscript to include these references, clearly distinguishing FuncBenchGen’s focus on controllable and contamination-free evaluation from the goals of prior works. This adequately contextualizes the paper’s contribution within the broader landscape.

**Reviewer Scores:**

* **Reviewer ncD9 (Score 4 -> 4 or 6):** The reviewer's concerns about formatting and examples were fully fixed. The "realism" concern was addressed with data, though the reviewer may still view the synthetic nature as a limitation.
* **Reviewer UX5z (Score 4 -> 4 or 6):** The technical questions about graph generation were answered clearly. Like ncD9, their score increase depends on their acceptance of the "synthetic proxy" argument.
* **Reviewer Hw9C (Score 4 -> 6):** The request for real-world comparison was directly met with the ToolHaystack/PaperArena data. This significantly strengthens the paper against their primary critique.
* **Reviewer cs9L (Score 6 -> 6 or 8):** This reviewer was already positive. The inclusion of missing literature and failure traces addresses their minor complaints, likely solidifying their support.
* **Reviewer Lq7b (Score 8 -> 8):** The reviewer was already a strong accept and the additional experiments only reinforce their position.

---

### Decision · Program_Chairs · 2026-01-26

Accept (Poster)